# Design, synthesis, and biological evaluation of phenylisoxazole-based histone deacetylase inhibitors

Xiaofei Qin[1,2,3,4,5]*, Meng Han[1,2,3,4,5], Peng Hu[1,2,3,4,5], Huadong Que[1,2,3,4,5], Zhenlei Shao[1,2,3,4,5], Dong Yan[1,2,3,4,5]

1 Department of Clinical Pharmacy, the First Affiliated Hospital of Xinxiang Medical University, Weihui, China, 2 Clinical Pharmacy, Henan Province Key Subjects of Medicine, the First Affiliated Hospital of Xinxiang Medical University, Weihui, China, 3 Xinxiang Key Laboratory for Individualized Drug Use Research for Immune Diseases, Weihui, China, 4 Xinxiang Anti-Oesophageal Cancer Nano-targeted Drug Delivery Engineering and Technology Research Centre, the First Affiliated Hospital of Xinxiang Medical University, Weihui, China, 5 Clinical Pharmacy, Xinxiang City Key Subjects of Medicine, the First Affiliated Hospital of Xinxiang Medical University, Weihui, China

* 18537333007@163.com

## Abstract

Histone deacetylases (HDACs) mediate the removal of acetyl groups from lysine residues in both histone and non-histone proteins, and have been regarded as promising targets for drug discovery. As a central member of HDAC family, HDAC1 has been found to be closely linked to the occurrence and development of prostate cancer. In this study, we designed and synthesized a new series of 3-phenylisoxazole HDAC1 inhibitors based on the hit **7**, identified by in-house compound library screening. The structure-activity relationship studies (SARs) indicated that the $R_1$ position was relatively tolerated for activity. The linker length at $R_2$ exerted a significant influence on activity with the relative orders of butyl＞propyl＞ethyl＞methyl. Among synthetic 16 compounds, compound **17** indicated the strongest HDAC1 inhibitory effect with the inhibition rate of 86.78% at the concentration of 1000 nM. In addition, derivative **17** could not only well occupy the active pocket of HDAC1, but also showed favorable drug-like properties. More importantly, molecule **17** exerted potent anti-proliferative activity on prostate cancer PC3 cells with the $IC_{50}$ value of 5.82 μM, and had no significant toxicity against normal prostate WPMY-1 cells. Collectively, these findings validate phenylisoxazole derivative **17** as a promising lead compound for further optimization and development.

## 1. Introduction

Histone modification serves as a pivotal epigenetic mechanism, particularly manifesting through the acetylation-deacetylation dynamics governed by histone acetyltransferases (HATs) and histone deacetylases (HDACs). HDACs predominantly

**Data availability statement:** All relevant data are within the paper and its Supporting Information files.

**Funding:** The author(s) received no specific funding for this work.

**Competing interests:** The authors have declared that no competing interests exist.

mediate the removal of acetyl groups from lysine residues in both histone and non-histone proteins, thereby inducing chromatin remodeling and orchestrating the transcriptional modulation of critical apoptotic and cell cycle regulatory genes [1–3]. Eleven zinc-dependent HDAC isoforms have been found in humans, phylogenetically divided into four classes based on structural domains, localization patterns, and homology. These classes consist of Class I (HDAC1/2/3/8), Class IIa (HDAC4/5/7/9), Class IIb (HDAC6/10), and Class IV (HDAC11) [4,5]. HDAC1 is usually checked in different organs. During the mitotic phase, HDAC1 protein is critical for condensing of chromatin, separating of chromosomes, and formatting of spindles [6]. Additionally, HDAC1 participates in multiple biological processes, such as red blood cell production, liver regeneration, programmed cell death, formation of new blood vessels, and cell division regulation [6]. HDACs establish dysregulation in many human diseases and are recognized as important therapeutic targets for cancers [3,7], diabetes [8], inflammatory processes [9,10], and cardiac diseases [11,12] and so on. In particular, the overexpression of HDAC1 plays a significant role in the progression of prostate cancer and is linked to a poor prognosis [13,14]. The expression level of HDAC1 is positively correlated with the abnormal proliferation of prostate cancer PC3 cells, which could be significantly reversed on treatment with an HDAC inhibitor TSA [15,16]. Moreover, HDAC1 downregulation results in E-cadherin expression and following inhibition of cell motility and invasion [17]. Targeting HDAC1 has been regarded as a promising approach for the treatment of prostate cancer.

Recently, the development of HDAC inhibitors (HDACIs) has emerged as a new strategy for innovative drug discovery. HDACIs design typically follows a pharmacophore model composed of three core elements: (i) a cap group that interacts with the entrance of catalytic center, (ii) a linker domain linking the cap to zinc-binding group (ZBG), and (iii) the ZBG moiety itself [18]. To date, five HDAC inhibitors (HDACIs), including Vorinostat (SAHA), Romidepsin, Tucidinostat, Panobinostat, and Belinostat, have been approved for use in hematologic cancers, while many clinical trials are currently assessing their effects in solid tumors (**Fig 1**) [18,19]. However, significant adverse effects associated with these approved HDACIs have been gradually discovered, including myelosuppression, gastrointestinal, cardiotoxicity, and hepatic abnormalities etc [20,21]. Therefore, the discovery of new HDAC inhibitors with low toxicity still holds significant importance and value.

We previously carried out an in-house compound library screening, providing a 3-phenylisoxazole derivative **7**, which showed 9.30% inhibitory activity against HDAC1 at 1000 nM (**Fig 2C**). Based on compound **7,** we herein performed further structural optimizations and structure-activity relationship studies, yielding a new series of 3-phenylisoxazole analogues, of which molecule **17** showed strong HDAC1 inhibitory activity, potent antiproliferative activity, good drug-like properties and low toxicity. Overall, this study offered a new structural skeleton for the development of drugs targeting HDACs.

**Fig 1. The structures of representative HDAC inhibitors.**

## 2. Results and discussion

### 2.1. Structure-based drug design of new 3-phenylisoxazole HDAC inhibitors

To discover new HDAC inhibitors, we conducted an in-house compound library screening, offering hit **7**, which indicated 9.3% inhibitory effect against HDAC1 at 1000 nM (**Fig 2C**). The molecular docking study shows that the amino hydrogen of hit **7** forms a π-H interaction with the benzene ring of Phe205. The hydroxyl hydrogen in the compound **7** has a hydrogen bond interaction with His141. In addition, the carbonyl oxygen in compound **7** demonstrates a relatively strong zinc-binding ability. Besides, the 4-chlorobenzyl group sits in a hydrophobic pocket. Based on these binding modes, structural optimizations and SAR studies were performed *via* focusing on $R_1$ and $R_2$ position (**Fig 2C**), in hope of further improving the hydrophobic interactions and zinc-binding ability.

### 2.2. Chemistry

The overall synthetic routes for new 3-phenylisoxazole derivatives were illustrated in **Fig 3**. Condensation of commercially available **1a-e** with hydroxylamine yielded intermediates **2a-e**, which were further chlorinated using *N*-chlorosuccinimide, resulted in the production of intermediates **3a-e**. intermediates **3a-e** were used to react with methyl 3-cyclopropyl-3-oxopropionate in the presence of triethylamine, providing cyclization intermediates **4a-e**. Hydrolysis of intermediates **4a-e** with sodium hydroxide produced key intermediates **5a-e**, which were condensed with different substituted amines, providing intermediates **6a-p**. Ammonia hydrolysis of intermediates **6a-p** with hydroxylamine under sodium hydroxide catalysis generated title compounds **7–16**.

### 2.3. Structure-activity relationship (SAR) studies

The inhibitory effects of compounds against HDAC1 were expressed in terms of inhibition rates at a single concentration. Vorinostat (SAHA), an approved HDAC inhibitor, was served as the positive control [22]. As shown in **Table 1**, compound **7** tethering a methyl group at $R_2$ position suppressed HDAC1 with the inhibition rate of 9.30% at the concentration of 1000 nM. The extension of methyl group in compound **7** with longer linkers resulted in compounds **8–11**. We observed that the linker length at $R_2$ position exerted a significant influence on activity with the relative orders of butyl group > propyl group > ethyl group > methyl group (**10** vs. **9** vs. **8** vs. **7**). In particular, molecule **10** indicated the strongest binding affinity

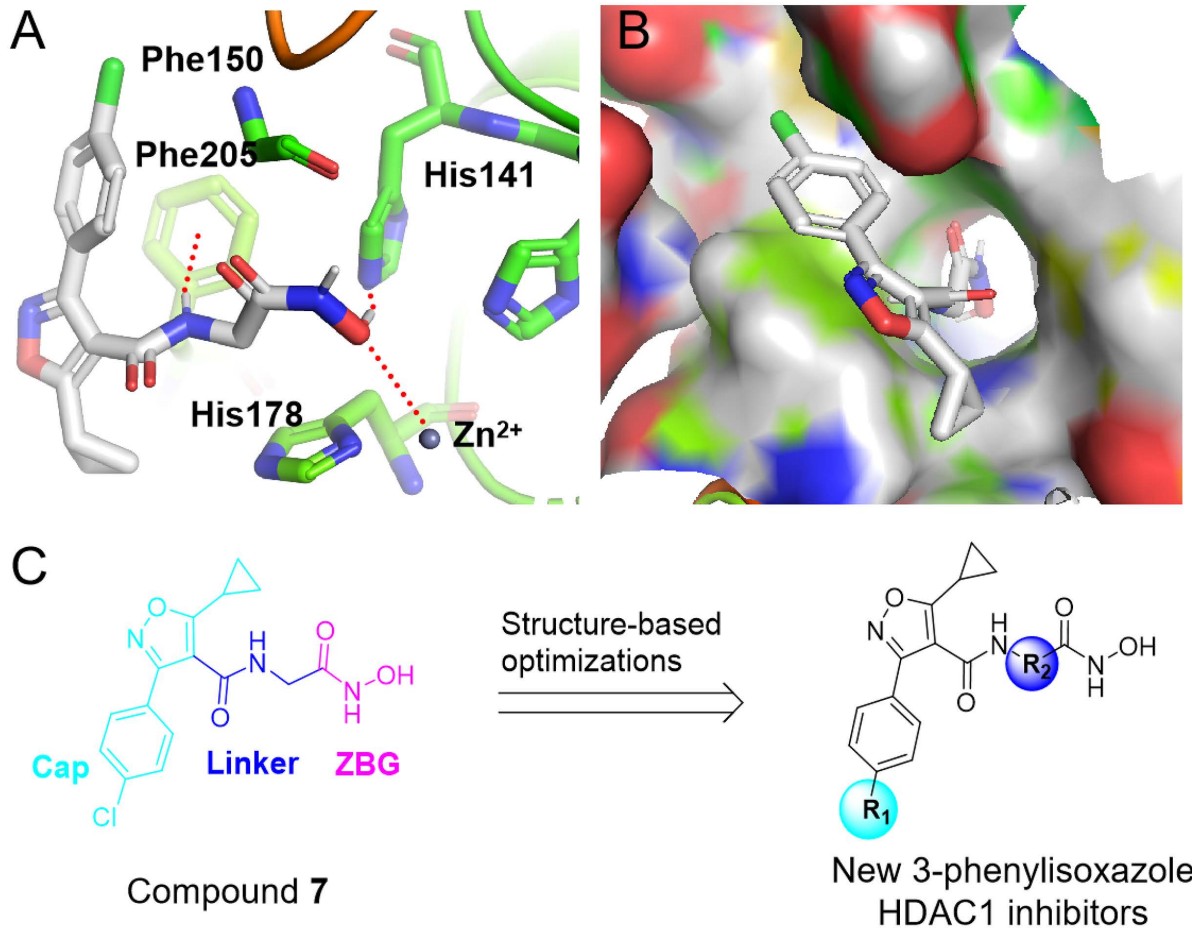

**Fig 2. The design strategy of new 3-phenylisoxazole HDAC1 inhibitors. (A-B)** The predicted binding modes of hit **7** (white) with HDAC1 (PDB: 5ICN). **(C)** The design scheme of new 3-phenylisoxazole HDAC1 inhibitors.

on HDAC1 with the inhibition rate of 79.85%, about 6-folds more potent than that of compound **7**. Replacement of butyl group in compound **10** with a benzyl group provided derivative **11**, which had the inhibition rate of 63.38%, slightly weaker activity in comparison to that of analogue **10**. Changing the chlorine atom of compounds **7–11** to methoxy, or cyano group resulted in compounds **14–22**, which exhibited similar activity trend. Next, the SARs studies of $R_1$ position were further conducted. As demonstrated in **Table 1**, compound **9** with a chlorine atom at $R_1$ position displayed the inhibition rate of 13.43%. However, translating the chlorine atom of compound **9** to other substituents, such as methyl (**12**), fluorine (**13**), methoxy (**16**) or cyano (**21**), did not resulted in an obvious alteration of HDAC1 inhibitory activity, suggesting that the $R_1$ position was relatively tolerated for activity. The SARs summary was depicted in **Fig 4**.

## 2.4. Drug-likeness studies of 3-phenylisoxazole derivatives 10, 17 and 22

Giving the potent inhibitory effects of compounds **10**, **17** and **22** towards HDAC1, we further assessed their drug-like properties through the admet SAR 3.0 website (https://lmmd.ecust.edu.cn/admetsar3/predict_submit.php). As indicated in **Table 2**, molecule **17** had a molecular weight of 373.41, number of atoms of 27, number of heteroatoms of 8, number of rings of 3, number of rotatable bonds of 9, number of hydrogen bond acceptors of 6, number of hydrogen bond donors of 3, topological polar surface area of 113.69, and the logarithm of n-octanol/water distribution coefficient

**Fig 3. The Synthetic route of 3-phenylisoxazole derivatives.** Reagents and conditions: **(a)** Hydroxylamine, EtOH, 60°C, 2 h; **(b)** N-Chlorosuccinimide, DMF, 40°C, 2 h; **(c)** Methyl 3-cyclopropyl-3-oxopropionate, triethylamine, EtOH, 0°C-re, 5-6 h; **(d)** NaOH, $H_2O$, 80°C, re, 1 h; **(e)** Suitable amine compounds, DIPEA, EDCI, rt, 2 h; **(f)** Hydroxylamine, NaOH, $H_2O$, rt, 1 **h.**

**Table 1. The HDAC1 inhibitory activity of 3-phenylisoxazole derivatives 6-20.**

| Compd. | $R_1$ | $R_2$ | HDAC1 inhibition 1000 nM (%) [a] |
|---|---|---|---|
| 7 | -Cl | -CH$_2$- | 9.30 |
| 8 | -Cl | -CH$_2$CH$_2$- | 11.84 |
| 9 | -Cl | -CH$_2$CH$_2$CH$_2$- | 13.43 |
| 10 | -Cl | -CH$_2$CH$_2$CH$_2$CH$_2$- | 79.85 |
| 11 | -Cl | −4-CH$_2$-phenyl- | 63.38 |
| 12 | -CH$_3$ | -CH$_2$CH$_2$CH$_2$- | 15.27 |
| 13 | -F | -CH$_2$CH$_2$CH$_2$- | 9.47 |
| 14 | -OCH$_3$ | -CH$_2$- | 9.93 |
| 15 | -OCH$_3$ | -CH$_2$CH$_2$- | 14.96 |
| 16 | -OCH$_3$ | -CH$_2$CH$_2$CH$_2$- | 17.37 |
| 17 | -OCH$_3$ | -CH$_2$CH$_2$CH$_2$CH$_2$- | 86.78 |
| 18 | -OCH$_3$ | −4-CH$_2$-phenyl- | 40.26 |
| 19 | -CN | -CH$_2$- | 11.46 |
| 20 | -CN | -CH$_2$CH$_2$- | 20.26 |
| 21 | -CN | -CH$_2$CH$_2$CH$_2$- | 28.15 |
| 22 | -CN | -CH$_2$CH$_2$CH$_2$CH$_2$- | 67.34 |
| Vorinostat | − | − | IC$_{50}$ = 12.79 ± 0.88 nM |

[a]Inhibition rate is presented as the mean derived from three independent determination.

**R$_1$ position was relatively tolerated for activity.**

**Linker length at R$_2$ exerted a significant influence on activity with the relative orders of butyl > propyl > ethyl > methyl**

**Fig 4. The summary for SAR studies of 3-phenylisoxazole derivatives.**

of 2.63. These properties made compound **17** fully satisfy both lipinski rule and pfizer rule (**Table 2**), which were generally applied to estimate the physicochemical properties and drug-like property of drugs. Meanwhile, molecules **10** and **22** also displayed similar drug-like properties (**Table 2**). Besides, compounds **10**, **17** and **22** indicated almost no significant effect on hERG, suggesting that these molecules probably did not cardiotoxicity. The analysis presented in **Fig 5**, generated by the ADMET SAR 3.0 website, further substantiates that these 3-phenylisoxazole derivatives possessed favorable drug-like properties.

**Table 2. The predicted drug-like parameters of representative compounds.**

| Basic Property Display | 10 | 17 | 22 |
|---|---|---|---|
| Lipinski Rule | Accept | Accept | Accept |
| Pfizer Rule | Accept | Accept | Accept |
| Molecular Weight | 377.83 | 373.41 | 368.39 |
| nAtom | 26 | 27 | 27 |
| nHet | 8 | 8 | 8 |
| nRing | 3 | 3 | 3 |
| nRot | 8 | 9 | 8 |
| HBA | 5 | 6 | 6 |
| HBD | 3 | 3 | 3 |
| TPSA | 104.46 | 113.69 | 128.25 |
| SlogP | 3.28 | 2.63 | 2.50 |
| hERG 1μM | 1.2% | 1.2% | 0.7% |

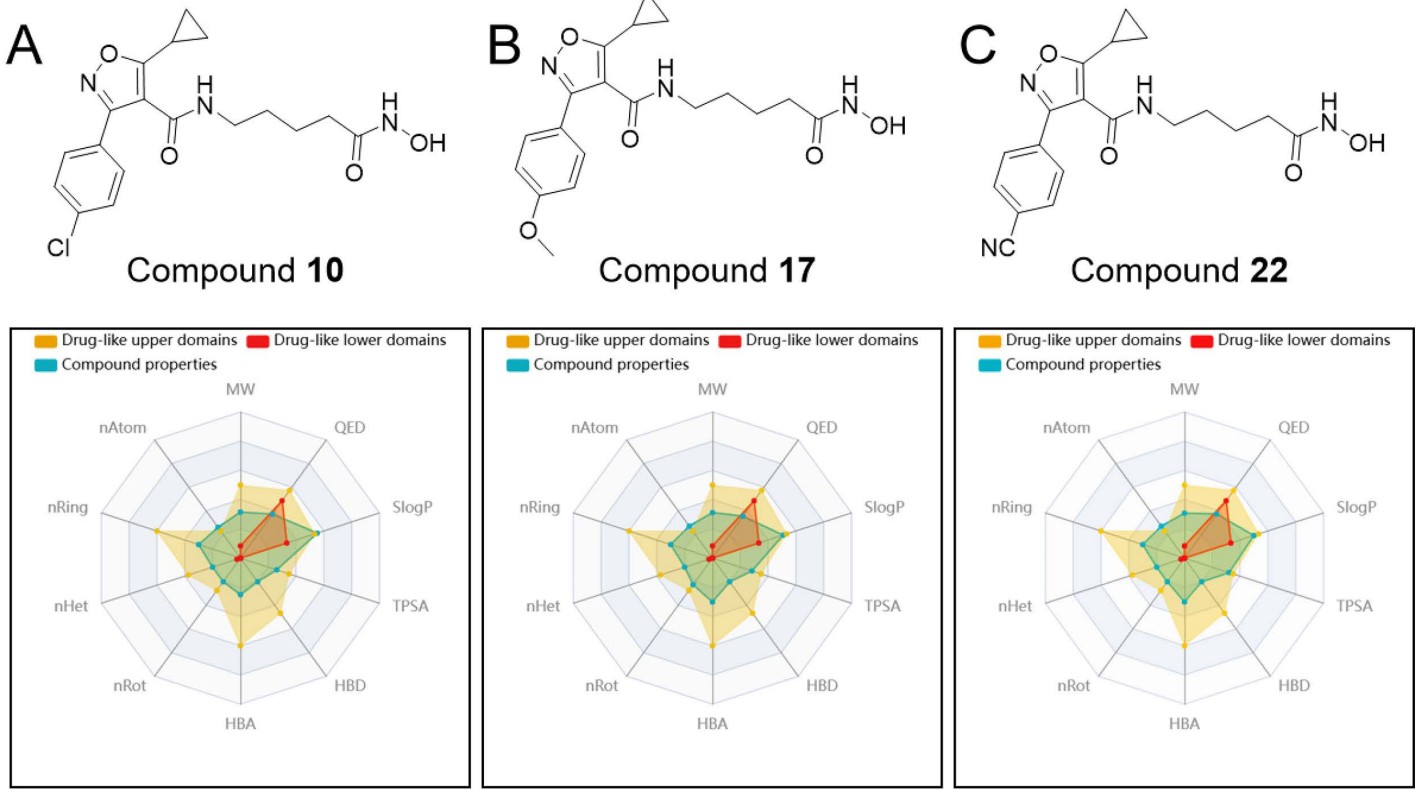

**Fig 5. The drug-like analysis diagrams of the compounds 10, 17 and 22.**

## 2.5. Anti-proliferative activity of representative 3-phenylisoxazole derivatives

Considering the potential therapeutic efficacy of HDAC1 inhibitors for prostate cancer, PC3 cells were used to evaluate *in vitro* anticancer activity of representative 3-phenylisoxazole derivatives **10** and **17**. The results showed that molecule

**10** inhibited PC3 cells with the IC$_{50}$ value of 9.18 μM (**Table 3**), slightly weaker in comparison to that of compound **17** (IC$_{50}$ = 5.82 μM). Interestingly, both compounds indicated no evident toxicity on prostate normal WPMY-1 cells. These results displayed that both compounds possessed potent anti-prostate cancer activity and low toxicity.

### 2.6. Molecular docking study

Encouraged by the potent activity, weak toxicity, and good drug-like properties of compound **17**, we next explored its binding modes with HDAC1 protein. As shown in **Figs 2** and **6**, compared to compound **7**, molecule **17** can better occupy the activity pocket of HDAC1. The amino hydrogen in compound **17** has a hydrogen bond interaction Asp99 and Gly149, respectively. Another hydrogen bond is formed between hydroxyl hydrogen of derivative **17** and imidazole nitrogen atom. Besides, the carbonyl oxygen of compound **17** is observed to have a strong zinc-binding effect. These interactions may be responsible for the high activity on HDAC1 of compound **17**.

### 3. Conclusion

In summary, a new class of 3-phenylisoxazole derivatives were designed and synthesized as potent HDAC1 inhibitors. The SARs exhibited that the R$_1$ position was well tolerated for activity. The linker length at R$_2$ showed a significant influence on activity with the relative orders of butyl > propyl > ethyl > methyl. The representative compound **17** displayed significant inhibitory potency against HDAC1 with the inhibition rate of 86.78% at the concentration of 1000 nM. MTT assay showed that derivative **17** possessed strong cytotoxicity toward prostate cancer PC3 cells, yet had no obvious influence on the growth of normal WPMY-1 cells. The docking study presented that derivative **17** could be well matched with the

**Table 3. The anti-proliferative activity of representative 3-phenylisoxazole derivatives.**

| Compd. | IC$_{50}$ (μM) [a] | |
|---|---|---|
| | PC3 | WPMY-1 |
| 10 | 9.18 ± 0.96 | > 40 |
| 17 | 5.82 ± 0.76 | > 40 |

[a]IC$_{50}$ value is presented as the mean derived from three independent determination.

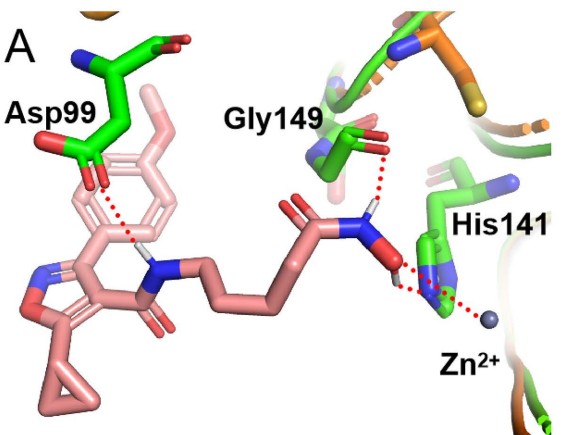
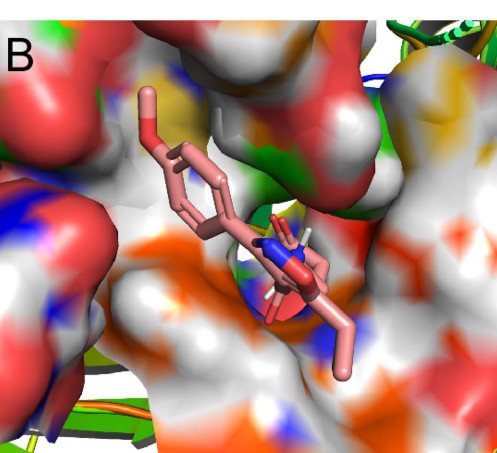

**Fig 6. The binding modes of compound 17 with HDAC1. (A)** Interactions between **17** (pink) and residues of HDAC1. **(B)** The binding pocket surface of HDAC1 and **17** (PDB: 5ICN).

active pockets of HDAC1. Besides, molecule **17** exhibited good drug-like properties. Overall, phenylisoxazole derivative **17** could serve as a lead compound for further optimizations in the treatment of prostate cancer.

## 4. Experimental section

### 4.1. General

Melting points were determined using a WRS-1A digital melting point apparatus. Thin-layer chromatography (TLC) was performed on silica gel-precoated glass plates with visualization under UV light (254 nm or 365 nm). All reagents and solvents were commercially sourced and used as received. Mass spectrometry data were acquired on a Waters Acquity UPLC system operating in positive electrospray ionization (ESI$^+$) mode. NMR spectra ($^1$H: 400 MHz; $^{13}$C: 100 MHz) were recorded on a Bruker 400 MHz spectrometer using DMSO-$d_6$ as the solvent.

### 4.2. Synthetic procedure compounds 2a-e

A mixture of commercially available benzaldehydes **1a-e** (1 equiv) and 50% hydroxylamine solution (2 equiv) in ethanol was refluxed at 60 °C for 2 h. After cooling to room temperature, the precipitated solid was filtered off and washed with ethanol to offer derivatives **2a-e** as white solid.

### 4.3. Synthetic procedure compounds 3a-e

A mixture of derivatives **2a-e** (1 equiv) and N-Chlorosuccinimide (5 equiv) in DMF was refluxed at 40 °C for 2 h. Upon completion of the reaction indicated by TLC, the mixture was quenched by water, and subsequently was extracted with ethyl acetate for three times. The combined organic layers were evacuated to provide the residue, which were recrystallized with ethyl acetate, producing title compounds **3a-e**.

### 4.4. Synthetic procedure compounds 4a-e

A mixture of derivatives **4a-e** (1 equiv), triethylamine (2 equiv) and methyl 3-cyclopropyl-3-oxopropionate (1 equiv) in ethanol was refluxed at room temperature for 5–6 h. Upon completion of the reaction indicated by TLC, the mixture was quenched by water, and subsequently was extracted with ethyl acetate for three times. The combined organic layers were evacuated to provide the residue, which were recrystallized with ethyl acetate, producing title compounds **4a-e**.

### 4.5. Synthetic procedure compounds 5a-e

A mixture of derivatives **4a-e** (1 equiv) and sodium hydroxide (3 equiv) in water was refluxed at 80 °C for 1 h. Upon completion of the reaction indicated by TLC, the resulting mixture was acidified with 0.5 N aq. HCl to pH 7–8. The precipitate was filtered, washed with water, recrystallized with ethanol and dried under reduced pressure to provide derivatives **5a-e**.

### 4.6. Synthetic procedure compounds 6a-p

A solution of **5a-e** (1 equiv), different substituted amines (1 equiv), N,N-diisopropylethylamine (1 equiv), and EDCI (1 equiv) in dichloromethane was reacted at room temperature for 2 h. Upon completion of the reaction indicated by TLC, the organic layers were evacuated to offer the residue, which was subsequently purified by chromatography (silica gel, 5% MeOH/DCM for **6a-f**, 6% MeOH/DCM for **6g**, 8% MeOH/DCM for **6h-l**, 7% MeOH/DCM for **6m-p**,) to offer compounds **6a-p**.

### 4.7. Synthetic procedure compounds 7–22

A solution of compounds **6a-p** (1 equiv), sodium hydroxide (1 equiv), and 50% hydroxylamine solution (2 equiv) in methanol were reacted at room temperature for 1 h. Upon completion of the reaction indicated by TLC, the resulting mixture was

evacuated to offer the residue, which was acidified with 0.5 N aq. formic acid to pH 6–7. The precipitate was subjected to filtration, followed by washing with water. It was then recrystallized using methanol and subsequently dried under reduced pressure, resulting in the formation of title compounds **7–22**.

3-(4-chlorophenyl)-5-cyclopropyl-N-(2-(hydroxyamino)-2-oxoethyl)isoxazole-4-carboxamide **(7)**. Yield 68%. White solid. Mp: 156.6–157.5 °C. $^1$H NMR (400 MHz, DMSO-$d_6$) δ 8.80 (t, $J$=6.0 Hz, 1H), 8.49 (s, 1H), 7.77 (d, $J$=8.2 Hz, 2H), 7.56 (d, $J$=8.1 Hz, 2H), 3.78 (d, $J$=5.8 Hz, 2H), 1.28–1.01 (m, 5H). $^{13}$C NMR (100 MHz, DMSO-$d_6$) δ 173.85, 166.21 (d, $J$=58.3 Hz), 162.22, 159.73, 135.30, 129.77 (d, $J$=92.0 Hz), 127.43, 112.26, 9.21, 8.36. MS (ESI), calcd. $C_{15}H_{14}ClN_3O_4$, [M+H] + m/z: 338.41. found: 338.42.

3-(4-chlorophenyl)-5-cyclopropyl-N-(3-(hydroxyamino)-3-oxopropyl)isoxazole-4-carboxamide **(8)**. Yield 65%. White solid. Mp: 177.4–178.2 °C. $^1$H NMR (400 MHz, DMSO-$d_6$) δ 10.53 (s, 1H), 8.88 (s, 1H), 8.58 (t, $J$=5.7 Hz, 1H), 7.68 (d, $J$=8.2 Hz, 2H), 7.59 (d, $J$=8.3 Hz, 2H), 3.42 (q, $J$=6.6 Hz, 2H), 2.33 (td, $J$=8.6, 4.4 Hz, 1H), 2.25 (t, $J$=7.0 Hz, 2H), 1.15 (dq, $J$=9.8, 6.0, 4.5 Hz, 2H), 1.09 (dd, $J$=5.2, 2.2 Hz, 2H). $^{13}$C NMR (100 MHz, DMSO-$d_6$) δ 173.59, 167.46, 161.78, 159.66, 135.31, 129.72 (d, $J$=60.3 Hz), 127.49, 112.61, 36.33, 32.44, 8.99, 8.28. MS (ESI), calcd. $C_{16}H_{16}ClN_3O_4$, [M+H] + m/z: 350.21. found: 350.22.

3-(4-chlorophenyl)-5-cyclopropyl-N-(4-(hydroxyamino)-4-oxobutyl) isoxazole-4-carboxamide **(9)**. Yield 66%. White solid. Mp: 136.8–137.6 °C. $^1$H NMR (400 MHz, DMSO-$d_6$) δ 10.39 (s, 1H), 8.51 (t, $J$=5.5 Hz, 1H), 7.68 (d, $J$=8.3 Hz, 2H), 7.59 (d, $J$=8.2 Hz, 2H), 3.21 (q, $J$=6.5 Hz, 2H), 2.45–2.27 (m, 1H), 2.12 (s, 1H), 2.01 (t, $J$=7.5 Hz, 1H), 1.71 (t, $J$=7.3 Hz, 2H), 1.17 (dd, $J$=8.0, 4.5 Hz, 2H), 1.09 (dt, $J$=8.4, 3.7 Hz, 2H). $^{13}$C NMR (100 MHz, DMSO-$d_6$) δ 173.35, 169.17, 161.74, 159.68, 135.36, 129.70 (d, $J$=57.0 Hz), 127.55, 112.77, 30.41, 25.51, 8.59 (d, $J$=62.2 Hz). MS (ESI), calcd. $C_{17}H_{18}ClN_3O_4$, [M+Na] + m/z: 386.19. found: 386.20.

3-(4-chlorophenyl)-5-cyclopropyl-N-(5-(hydroxyamino)-5-oxopentyl) isoxazole-4-carboxamide **(10)**. Yield 67%. White solid. Mp: 124.3–125.7 °C. $^1$H NMR (400 MHz, DMSO-$d_6$) δ 10.38 (s, 1H), 8.51 (t, $J$=5.8 Hz, 1H), 7.68 (d, $J$=8.2 Hz, 2H), 7.60 (d, $J$=8.1 Hz, 2H), 3.34 (s, 2H), 3.21 (q, $J$=6.3 Hz, 2H), 2.29 (tt, $J$=8.4, 4.9 Hz, 1H), 1.98 (t, $J$=7.0 Hz, 1H), 1.49 (dq, $J$=22.0, 7.6 Hz, 4H), 1.17 (dq, $J$=7.6, 4.2 Hz, 2H), 1.09 (dq, $J$=7.8, 4.9 Hz, 2H). $^{13}$C NMR (100 MHz, DMSO-$d_6$) δ 173.22, 169.38, 161.69, 159.65, 135.36, 129.68 (d, $J$=49.6 Hz), 127.56, 112.87, 32.34, 28.87, 23.12, 8.84, 8.27. MS (ESI), calcd. $C_{18}H_{20}ClN_3O_4$, [M+Na] + m/z: 400.17. found: 400.16.

3-(4-chlorophenyl)-5-cyclopropyl-N-(4-(hydroxycarbamoyl) benzyl) isoxazole-4-carboxamide **(11)**. Yield 63%. White solid. Mp: 210.5–211.7 °C. $^1$H NMR (400 MHz, DMSO-$d_6$) δ 9.03 (t, $J$=6.0 Hz, 1H), 7.74 (d, $J$=8.1 Hz, 2H), 7.62 (d, $J$=8.5 Hz, 2H), 7.58–7.45 (m, 2H), 7.35 (d, $J$=8.0 Hz, 2H), 4.46 (d, $J$=5.9 Hz, 2H), 2.32 (s, 1H), 1.17 (dq, $J$=8.3, 3.3, 2.8 Hz, 2H), 1.10 (dt, $J$=5.6, 3.0 Hz, 2H). $^{13}$C NMR (100 MHz, DMSO-$d_6$) δ 173.57, 164.33, 161.93, 159.79, 142.39, 135.37, 132.10, 130.06, 129.34, 127.88, 127.44 (d, $J$=3.6 Hz), 112.56, 42.99, 8.95, 8.31. MS (ESI), calcd. $C_{21}H_{18}ClN_3O_4$, [M+H] + m/z: 412.21. found: 412.20.

5-cyclopropyl-N-(4-(hydroxyamino)-4-oxobutyl)-3-(p-tolyl)isoxazole-4-carboxamide **(12)**. Yield 66%. White solid. Mp: 142.5–146.3 °C. $^1$H NMR (400 MHz, DMSO-$d_6$) δ 8.51 (d, $J$=5.8 Hz, 1H), 8.47 (d, $J$=15.6 Hz, 1H), 7.55 (d, $J$=7.8 Hz, 2H), 7.31 (d, $J$=7.8 Hz, 2H), 3.19 (q, $J$=6.7 Hz, 2H), 2.37 (s, 3H), 2.32–2.20 (m, 1H), 2.00 (t, $J$=7.5 Hz, 1H), 1.79–1.60 (m, 2H), 1.15 (dt, $J$=8.4, 3.2 Hz, 2H), 1.07 (dq, $J$=7.7, 4.9, 4.2 Hz, 2H). $^{13}$C NMR (100 MHz, DMSO-$d_6$) δ 172.74, 169.10, 166.35, 162.05, 160.42, 140.19, 129.81, 128.01, 125.85, 112.80, 30.41, 25.54, 21.43, 8.70, 8.22. MS (ESI), calcd. $C_{18}H_{21}N_3O_4$, [M+H] + m/z: 344.32. found: 344.31.

5-cyclopropyl-3-(4-fluorophenyl)-N-(4-(hydroxyamino)-4-oxobutyl)isoxazole-4-carboxamide **(13)**. Yield 62%. White solid. Mp: 115.3–116.2 °C. $^1$H NMR (400 MHz, DMSO-$d_6$) δ 8.51 (s, 1H), 8.40 (s, 1H), 7.66–7.56 (m, 2H), 7.06 (d, $J$=120.8 Hz, 2H), 3.83 (s, 3H), 2.33–2.19 (m, 1H), 2.01 (t, $J$=7.5 Hz, 1H), 1.78–1.66 (m, 2H), 1.14 (dt, $J$=8.3, 3.2 Hz, 2H), 1.07 (dt, $J$=5.6, 3.1 Hz, 2H). $^{13}$C NMR (100 MHz, DMSO-$d_6$) δ 172.62, 169.15, 165.86, 162.15, 161.06, 160.06, 157.49, 129.56, 120.92, 114.72, 112.67, 55.79, 30.42, 25.55, 8.67, 8.21. MS (ESI), calcd. $C_{17}H_{18}FN_3O_4$, [M+Na] + m/z: 370.24. found: 370.25.

 

5-cyclopropyl-N-(2-(hydroxyamino)-2-oxoethyl)-3-(4-methoxyphenyl)isoxazole-4-carboxamide **(14)**. Yield 65%. White solid. Mp: 149.7–150.2 °C. ¹H NMR (400 MHz, DMSO-$d_6$) δ 10.58 (s, 1H), 8.88 (s, 1H), 8.67 (t, $J=6.0$ Hz, 1H), 7.63 (d, $J=7.9$ Hz, 2H), 7.28 (d, $J=7.9$ Hz, 3H), 3.76 (d, $J=5.9$ Hz, 2H), 2.52 (p, $J=1.9$ Hz, 2H), 2.47 (d, $J=10.0$ Hz, 1H), 1.13 (ddt, $J=10.9, 7.9, 2.7$ Hz, 5H). ¹³C NMR (100 MHz, DMSO-$d_6$) δ 173.34, 166.04, 162.50, 160.48, 140.11, 129.77, 128.25, 125.72, 112.22, 21.44, 9.06, 8.30. MS (ESI), calcd. $C_{16}H_{17}N_3O_5$, [M + Na] + m/z: 354.23. found: 354.23/338.23.

5-cyclopropyl-N-(3-(hydroxyamino)-3-oxopropyl)-3-(4-methoxyphenyl)isoxazole-4-carboxamide **(15)**. Yield 63%. White solid. Mp: 174.5–175.2 °C. ¹H NMR (400 MHz, DMSO-$d_6$) δ 8.53 (t, $J=5.7$ Hz, 1H), 8.42 (s, 1H), 7.80 (s, 1H), 7.54 (d, $J=7.9$ Hz, 2H), 7.30 (d, $J=7.9$ Hz, 2H), 3.41 (q, $J=6.7$ Hz, 2H), 2.52 (p, $J=1.8$ Hz, 2H), 2.36 (s, 1H), 2.23 (d, $J=7.2$ Hz, 2H), 1.24–1.10 (m, 2H), 1.06 (dq, $J=7.7, 5.0, 4.3$ Hz, 2H). ¹³C NMR (100 MHz, DMSO-$d_6$) δ 172.98, 167.40, 166.07, 162.07, 160.44, 157.46, 140.12, 129.82, 128.08, 125.80, 112.65, 36.34, 32.49, 21.44, 8.82, 8.22. MS (ESI), calcd. $C_{17}H_{19}N_3O_5$, [M + Na] + m/z: 368.23. found: 352.23/368.23.

5-cyclopropyl-N-(4-(hydroxyamino)-4-oxobutyl)-3-(4- methoxyphenyl) isoxazole-4-carboxamide **(16)**. Yield 65%. White solid. Mp: 158.8–159.7 °C. ¹H NMR (400 MHz, DMSO-$d_6$) δ 8.50 (t, $J=5.7$ Hz, 1H), 8.40 (d, $J=1.8$ Hz, 1H), 7.67–7.49 (m, 2H), 7.17–6.82 (m, 2H), 3.82 (d, $J=1.4$ Hz, 3H), 3.20 (q, $J=6.8$ Hz, 2H), 2.36–2.17 (m, 1H), 2.01 (t, $J=7.5$ Hz, 2H), 1.71 (q, $J=7.4$ Hz, 2H), 1.24–1.14 (m, 2H), 1.06 (dt, $J=5.3, 3.3$ Hz, 2H). ¹³C NMR (100 MHz, DMSO-$d_6$) δ 172.62, 169.15, 165.87, 162.15, 160.56 (d, $J=100.8$ Hz), 157.49, 129.55, 120.92, 113.69 (d, $J=206.5$ Hz), 55.78, 30.42, 25.55, 8.66, 8.21. MS (ESI), calcd. $C_{18}H_{21}N_3O_5$, [M + Na] + m/z: 382.14. found: 382.15.

5-cyclopropyl-N-(5-(hydroxyamino)-5-oxopentyl)-3-(4-methoxyphenyl)isoxazole-4-carboxamide **(17)**. Yield 68%. White solid. Mp: 142.5–143.7 °C. ¹H NMR (400 MHz, DMSO-$d_6$) δ 10.45 (s, 1H), 8.58–8.34 (m, 1H), 7.60 (d, $J=8.4$ Hz, 2H), 7.06 (d, $J=8.4$ Hz, 2H), 3.82 (s, 3H), 3.20 (q, $J=6.3$ Hz, 2H), 2.25 (dt, $J=8.4, 3.6$ Hz, 1H), 1.98 (t, $J=7.0$ Hz, 2H), 1.49 (dq, $J=21.6, 7.5$ Hz, 4H), 1.14 (dd, $J=7.9, 3.3$ Hz, 2H), 1.05 (dq, $J=7.7, 4.8, 4.3$ Hz, 2H). ¹³C NMR (100 MHz, DMSO-$d_6$) δ 172.52, 169.33, 166.62, 162.10, 161.06, 160.02, 129.51, 120.92, 114.73, 112.75, 55.78, 32.38, 28.89, 23.15, 8.63, 8.20. MS (ESI), calcd. $C_{19}H_{23}N_3O_5$, [M + Na] + m/z: 396.21. found: 396.20.

5-cyclopropyl-N-(4-(hydroxycarbamoyl)benzyl)-3-(4-methoxyphenyl)isoxazole-4-carboxamide **(18)**. Yield 63%. White solid. Mp: 205.5–206.3 °C. ¹H NMR (400 MHz, DMSO-$d_6$) δ 8.98 (s, 1H), 8.53 (s, 1H), 7.71 (d, $J=7.8$ Hz, 2H), 7.53 (d, $J=8.4$ Hz, 2H), 7.25 (d, $J=7.9$ Hz, 2H), 6.97 (d, $J=8.4$ Hz, 2H), 4.42 (d, $J=5.9$ Hz, 2H), 3.81 (s, 3H), 2.26 (s, 1H), 1.13 (dt, $J=8.5, 3.1$ Hz, 2H), 1.06 (q, $J=3.5, 2.6$ Hz, 2H). ¹³C NMR (100 MHz, DMSO-$d_6$) δ 166.67, 129.60, 129.55, 129.20, 127.73, 127.04, 114.63, 114.12, 55.78, 8.67 (d, $J=2.8$ Hz), 8.58, 8.22, 8.20. MS (ESI), calcd. $C_{22}H_{21}N_3O_5$, [M + H] + m/z: 408.15. found: 408.16.

3-(4-cyanophenyl)-5-cyclopropyl-N-(2-(hydroxyamino)-2-oxoethyl)isoxazole-4-carboxamide **(19)**. Yield 63%. White solid. Mp: 152.4–153.5 °C. ¹H NMR (400 MHz, DMSO-$d_6$) δ 8.82 (t, $J=6.2$ Hz, 1H), 8.48 (s, 1H), 7.96 (d, $J=8.1$ Hz, 1H), 7.77 (dq, $J=17.6, 8.4, 7.9$ Hz, 2H), 5.89 (s, 1H), 3.77 (d, $J=5.9$ Hz, 2H), 2.52–2.42 (m, 1H), 1.26–1.15 (m, 2H), 1.14–1.01 (m, 2H). ¹³C NMR (100 MHz, DMSO-$d_6$) δ 173.53, 166.34, 165.94, 162.39, 150.77, 135.24, 128.84, 128.32, 128.25, 128.11, 126.10, 9.18, 9.13, 8.34. MS (ESI), calcd. $C_{16}H_{14}N_4O_4$, [M + K] + m/z: 367.06. found: 367.16.

3-(4-cyanophenyl)-5-cyclopropyl-N-(3-(hydroxyamino)-3-oxopropyl)isoxazole-4-carboxamide **(20)**. Yield 67%. White solid. Mp: 173.2–174.5 °C. ¹H NMR (400 MHz, DMSO-$d_6$) δ 8.63 (q, $J=5.5, 4.9$ Hz, 1H), 8.49 (s, 1H), 7.98 (s, 1H), 7.90–7.81 (m, 1H), 7.81–7.67 (m, 1H), 7.65 (d, $J=8.1$ Hz, 1H), 5.90 (s, 1H), 3.43 (q, $J=6.8$ Hz, 2H), 2.42–2.27 (m, 1H), 2.25 (d, $J=7.3$ Hz, 2H), 1.17 (dt, $J=8.5, 3.8$ Hz, 2H), 1.08 (s, 2H). ¹³C NMR (100 MHz, DMSO-$d6$) δ 167.42, 166.46, 161.94, 150.80, 135.29, 128.20, 128.08, 127.95, 126.20, 112.71, 36.35, 32.48, 8.99, 8.92, 8.27. MS (ESI), calcd. $C_{17}H_{16}N_4O_4$, [M + Na] + m/z: 374.26. found: 374.25.

3-(4-cyanophenyl)-5-cyclopropyl-N-(4-(hydroxyamino)-4-oxobutyl)isoxazole-4-carboxamide **(21)**. Yield 63%. White solid. Mp: 182.4–183.3 °C. ¹H NMR (400 MHz, DMSO-$d_6$) δ 8.57 (t, $J=5.6$ Hz, 1H), 8.51 (d, $J=1.3$ Hz, 1H), 7.97 (s, 1H), 7.83 (d, $J=19.4$ Hz, 1H), 7.70 (dd, $J=17.6, 9.3$ Hz, 1H), 7.64 (s, 1H), 5.90 (s, 1H), 3.21 (q, $J=6.7$ Hz, 2H), 2.30 (td, $J=8.5, 4.3$ Hz, 1H), 2.08–1.94 (m, 2H), 1.70 (q, $J=7.4$ Hz, 2H), 1.17 (dq, $J=7.2, 3.7$ Hz, 2H), 1.09 (dt, $J=6.6, 3.0$ Hz, 2H). ¹³C

NMR (100 MHz, DMSO-d6) δ 169.11, 166.76, 161.78, 160.16, 160.07, 150.74, 128.36, 128.11, 128.02, 127.88, 127.69, 112.93, 30.42, 25.55, 8.87, 8.80, 8.26. MS (ESI), calcd. $C_{18}H_{18}N_4O_4$, [M+K] + m/z: 395.09. found: 395.23.

3-(4-cyanophenyl)-5-cyclopropyl-N-(5-(hydroxyamino)-5-oxopentyl)isoxazole-4-carboxamide **(22)**. Yield 67%. White solid. Mp: 208.0–209.5 °C. $^1$H NMR (400 MHz, DMSO-$d_6$) δ 10.36 (s, 1H), 8.69 (s, 1H), 8.50 (d, $J = 5.9$ Hz, 1H), 8.20–7.93 (m, 2H), 7.75 (dd, $J = 28.9, 8.0$ Hz, 2H), 3.21 (q, $J = 6.4$ Hz, 2H), 2.29 (dt, $J = 8.7, 3.5$ Hz, 1H), 1.98 (t, $J = 7.0$ Hz, 2H), 1.58–1.40 (m, 4H), 1.23–1.03 (m, 4H). $^{13}$C NMR (100 MHz, DMSO-$d_6$) δ 173.17, 168.61 (d, $J = 163.3$ Hz), 161.73, 160.02, 136.08, 131.23, 128.36, 127.99, 127.85, 126.18, 113.02, 32.34, 28.85, 23.12, 8.83, 8.26. MS (ESI), calcd. $C_{19}H_{20}N_4O_4$, [M+K] + m/z: 409.35. found: 409.34.

## 4.8. MTT assay

Cells were seeded in 96-well plates at a density of $2-3 \times 10^3$ cells per well. On the following day, 200 µL of fresh complete medium containing serial concentrations of tested compounds was added to the plates, which were then incubated for 24 h, 48 h, and 72 h individually. For PC3 and WPMY-1 cells, pre-treatment with 4 mM SB203580 was performed for 4 h, followed by treatment with the $IC_{50}$ concentration of the compounds for 72 h. Subsequently, 20 µL of MTT solution (5 mg/mL, dissolved in PBS) was added to each well. After an additional 4 h incubation at 37°C in the dark, absorbance was measured at 490 nm using a microplate reader. The absorbance values were analyzed, and $IC_{50}$ values were calculated using SPSS software.

## 4.9. HDAC1 inhibition assay

A 100 µL mixture of HDAC1 solution and compounds at different concentrations was incubated in assay buffer (2 mM $MgCl_2$; 2 mM KCl; 100 mM NaCl; 20 mM HEPES, pH 7.5) at 37°C for a duration of 2 minutes. Next, the substrate Boc-LGK-Acetyl-AMC was added to achieve a final concentration of 1 mM, and the reactions were allowed to proceed at 37°C for an additional 20 minutes. To terminate the reactions, trypsin was introduced, followed by a subsequent incubation period of 15 minutes at 37°C. Fluorescence intensity was determined with a microplate reader with excitation set to 355 nm and emission detected at 460 nm. Inhibitory ratios were calculated utilizing Prism version 6.0 software [23].

## 4.10. Molecular docking

Molecular docking studies were conducted utilizing the Molecular Operating Environment (MOE) software. The crystal structure of HDAC1 (PDB ID: 5ICN) was employed as the docking receptor [24]. Receptor preparation involved the addition of missing atoms, removal of water molecules, and energy minimization with the QuickPrep module. Three-dimensional structures for derivatives **7** and **17** were generated *via* conformational searching and subsequent energy minimization. The protonation states of both ligand and receptor were adjusted to a pH of 7.0. During the docking process, HDAC1 protein was maintained in a rigid conformation while allowing flexibility for small molecules. The top 20 scoring poses were selected for analysis. The binding mode visualizations were produced through PyMOL [25].

## Supporting information

**S1 File. Compounds 7–22 characterization.**
(DOCX)

## Acknowledgments

I want to thank Haomiao Jiao for her suggestions on the design of this project and the writing of the paper.

## Author contributions

**Data curation:** Xiaofei Qin.

**Formal analysis:** Peng Hu.

**Funding acquisition:** Xiaofei Qin.

**Investigation:** Peng Hu.

**Methodology:** Peng Hu, Zhenlei Shao.

**Resources:** Huadong Que, Dong Yan.

**Software:** Meng Han, Peng Hu, Huadong Que, Zhenlei Shao.

**Validation:** Xiaofei Qin, Huadong Que, Zhenlei Shao.

**Visualization:** Meng Han, Huadong Que.

**Writing – original draft:** Xiaofei Qin, Meng Han.

**Writing – review & editing:** Dong Yan.

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
