## [Decision Letter · Decision Letter 0]

27 Aug 2025

Dear Dr. Qin,

Thank you for submitting your manuscript to PLOS ONE. After careful consideration, we feel that it has merit but does not fully meet PLOS ONE’s publication criteria as it currently stands. Therefore, we invite you to submit a revised version of the manuscript that addresses the points raised during the review process.

If applicable, we recommend that you deposit your laboratory protocols in protocols.io to enhance the reproducibility of your results. Protocols.io assigns your protocol its own identifier (DOI) so that it can be cited independently in the future. For instructions see: https://journals.plos.org/plosone/s/submission-guidelines#loc-laboratory-protocols. Additionally, PLOS ONE offers an option for publishing peer-reviewed Lab Protocol articles, which describe protocols hosted on protocols.io. Read more information on sharing protocols at https://plos.org/protocols?utm_medium=editorial-email&utm_source=authorletters&utm_campaign=protocols .

We look forward to receiving your revised manuscript.

Kind regards,

Afzal Basha Shaik, Ph.D

Academic Editor

PLOS ONE

Journal Requirements:

4. Please note that your Data Availability Statement is currently missing [the repository name and/or the DOI/accession number of each dataset OR a direct link to access each database]. If your manuscript is accepted for publication, you will be asked to provide these details on a very short timeline. We therefore suggest that you provide this information now, though we will not hold up the peer review process if you are unable.

Reviewers' comments:

Reviewer's Responses to Questions

**Comments to the Author**

1. Is the manuscript technically sound, and do the data support the conclusions?

Reviewer #1: Yes

Reviewer #2: Partly

2. Has the statistical analysis been performed appropriately and rigorously?

Reviewer #1: Yes

Reviewer #2: No

3. Have the authors made all data underlying the findings in their manuscript fully available?

Reviewer #1: Yes

Reviewer #2: Yes

4. Is the manuscript presented in an intelligible fashion and written in standard English?

Reviewer #1: Yes

Reviewer #2: No

Reviewer #1: Manuscript Title: Design, synthesis, and biological evaluation of phenylisoxazole-based histone deacetylase inhibitors

Manuscript ID: PONE-D-25-33411

General Comments: The design, manufacturing, and pharmacological assessment of phenylisoxazole-based histone deacetylase (HDAC) inhibitors that specifically target HDAC1 are thoroughly examined in this publication. In silico modelling, synthetic chemistry, SAR analysis, drug-likeness assessment, and biological activity evaluation are all integrated in this well-structured work, which offers important insights into this unique scaffold.

Strengths and Significance:

1. Novel Scaffold Development: An important addition to the chemical diversity of HDAC inhibitors is the discovery and refinement of phenylisoxazole derivatives as HDAC1 inhibitors, which go beyond traditional hydroxamic acids and benzamides. Potential benefits in terms of toxicity profiles and selectivity are provided by the phenylisoxazole core.

2. Rational Design and SAR: The alterations at R1 and R2 positions were successfully directed by the structure-based drug design approach, which was aided by molecular docking. This resulted in compounds with increased activity, most notably compound 17. The comprehensive SAR analysis shows that linker length at R2 has a major impact on activity, with longer chains—like butyl—providing greater potency.

3. Synthetic Accessibility: Using widely accessible starting materials and conventional reactions, the synthetic approach outlined is simple and allows for additional optimisation and scalability.

4. Biological Evaluation: Potential therapeutic relevance is demonstrated by the combination of antiproliferative studies on prostate cancer PC3 cells and enzymatic HDAC1 inhibition assays. It is especially encouraging that WPMY-1 is selective for cancer cells rather than healthy prostate cells.

5. Drug-Likeness and ADMET Predictions: Compound 17 and its derivatives have good drug-like profiles, according to the computational assessment of pharmacokinetic characteristics and cardiotoxicity risk (hERG inhibition).

All things considered, this work presents a novel scaffold for HDAC1 inhibition that shows promise for treating prostate cancer. It is admirable how logical design, synthetic chemistry, and biological assessment have been combined. Future research addressing the aforementioned factors will enhance these compounds' translational potential and promote their advancement as medicinal agents.

Hence, I recommend this manuscript for publication.

Additionally, if possible the authors can add the below references. Thank you.

1. Cao, D., Zhou, X., Guo, Q., Xiang, M., Bao, M., He, B.,... Mao, X. (2024). Unveiling the role of histone deacetylases in neurological diseases: focus on epilepsy. Biomarker Research, 12(1), 142. doi: 10.1186/s40364-024-00687-6

2. Huang, T., Chen, Y., Zhao, Q., Wu, X., Li, H., Luo, X.,... Tang, N. (2025). Dual Regulation of Sprouty 4 Palmitoylation by ZDHHC7 and Palmitoyl-Protein Thioesterase 1: A Potential Therapeutic Strategy for Cisplatin-Resistant Osteosarcoma. Research, 8, 708. doi: 10.34133/research.0708

Reviewer #2: Manuscript Number: PONE-D-25-33411

Title: Design, synthesis, and biological evaluation of phenylisoxazole-based histone deacetylase inhibitors

The submitted manuscript describes the design, synthesis, and biological evaluation of phenylisoxazole-based histone deacetylase (HDAC) inhibitors as potential anticancer agents. While the topic aligns with the scope of the journal, the work presents limited novelty and requires substantial revisions before it can be reconsidered for publication. The authors are advised to address the following major concerns:

1. Please rewrite the keywords in alphabetical order to ensure consistency with journal formatting guidelines.

2. The manuscript contains multiple grammatical and typographical errors. A thorough language revision is necessary. Specifically:

a) Replace the word "novel" with "new" throughout the manuscript,

b) Ensure that all captions for schemes, tables, and figures are clearly indicated in bold within the main text.

c) Replace the abbreviation "eq" with "equiv." for consistency and clarity.

3. The HDAC1 inhibitory activity of Vorinostat (used as the positive control) should be included in Table 1 for comparison with the synthesized compounds.

4. There is inconsistency in the use of compound codes throughout the manuscript. Please review all compound identifiers carefully and present them in a uniform format, especially in schemes and Table 1.

5. The general procedure provided for the synthesis of compounds 6a–6p uses 5% DCM: MeOH as the eluent. However, considering the structural variation among these compounds, it is unlikely that a single eluent system would be adequate for the purification of all derivatives. It is recommended to describe each reaction step individually, including the exact quantities of reagents and detailed purification methods for each compound.

6. Please verify and correct the M/Z values for compounds 19, 21, and 22. Ensure that all reported mass data are accurate and correspond to the correct molecular formulas

**Do you want your identity to be public for this peer review?** For information about this choice, including consent withdrawal, please see our Privacy Policy

Reviewer #1: No

Reviewer #2: **Yes: ** Danaboina Srikanth

---

## [Author Response · Author response to Decision Letter 1]

16 Sep 2025

Responds to editor’s comments

Re: According to the PLOS ONE's style requirements, we have revised the manuscript.

Re: We have provided the complete Data Availability Statement on page 20 of the revised manuscript in red.

Re: https://orcid.org/0009-0009-4156-2708

4. Please note that your Data Availability Statement is currently missing [the repository name and/or the DOI/accession number of each dataset OR a direct link to access each database]. If your manuscript is accepted for publication, you will be asked to provide these details on a very short timeline. We therefore suggest that you provide this information now, though we will not hold up the peer review process if you are unable.

Re: Not applicable

Re: After review and evaluation, we cited a literature recommended by Reviewer 1.

Responds to reviewers’ comments

Responds to reviewer 1

1. Regarding the reference and citation of recommended literatures

Re: According to your suggestion, we have cited the recommended literatures.

Responds to reviewer 2

1. Please rewrite the keywords in alphabetical order to ensure consistency with journal formatting guidelines.

Re: According to your suggestion, we have rewritten the keywords in alphabetical order.

2. The manuscript contains multiple grammatical and typographical errors. A thorough language revision is necessary. Specifically:

a) Replace the word "novel" with "new" throughout the manuscript,

b) Ensure that all captions for schemes, tables, and figures are clearly indicated in bold within the main text.

c) Replace the abbreviation "eq" with "equiv." for consistency and clarity.

Re: According to your suggestion, we have carefully revised them in red.

3. The HDAC1 inhibitory activity of Vorinostat (used as the positive control) should be included in Table 1 for comparison with the synthesized compounds.

Re: According to your suggestion, we have added the HDAC1 inhibitory activity of Vorinostat to Table 1 in red.

4. There is inconsistency in the use of compound codes throughout the manuscript. Please review all compound identifiers carefully and present them in a uniform format, especially in schemes and Table 1.

Re: According to your suggestion, we have revised them.

5. The general procedure provided for the synthesis of compounds 6a–6p uses 5% DCM: MeOH as the eluent. However, considering the structural variation among these compounds, it is unlikely that a single eluent system would be adequate for the purification of all derivatives. It is recommended to describe each reaction step individually, including the exact quantities of reagents and detailed purification methods for each compound.

Re: As is well known, there are many factors influencing the purification conditions of column chromatography. Therefore, based on your suggestions and by referring to other literatures [1, 2], we can only provide the approximate purification conditions of the compounds as much as possible. We have revised the general procedures on page 14 of the revised manuscript in red.

6. Please verify and correct the M/Z values for compounds 19, 21, and 22. Ensure that all reported mass data are accurate and correspond to the correct molecular formulas

Re: According to your suggestion, we have corrected them on page 18-19 of the revised manuscript in red.

References

[1] X. Cao, P. Wang, W. Zhao, H. Yuan, H. Hu, T. Chen, Y. Zhang, Y. Ren, L. Su, K. Fu, H. Liu, D. Guo, Structure–Affinity and Structure–Kinetic Relationship Studies of Benzodiazepine Derivatives for the Development of Efficacious Vasopressin V2 Receptor Antagonists, Journal of Medicinal Chemistry 66(5) (2023) 3621-3634.

[2] H. Zhang, W. Yan, Y. Sun, H. Yuan, L. Su, X. Cao, P. Wang, Z. Xu, Y. Hu, Z. Wang, Y. Wang, K. Fu, Y. Sun, Y. Chen, J. Cheng, D. Guo, Long Residence Time at the Vasopressin V2 Receptor Translates into Superior Inhibitory Effects in Ex Vivo and In Vivo Models of Autosomal Dominant Polycystic Kidney Disease, Journal of Medicinal Chemistry 65(11) (2022) 7717-7728.

---

## [Editor Report · Decision Letter 1]

1 Oct 2025

Design, synthesis, and biological evaluation of phenylisoxazole-based histone deacetylase inhibitors

PONE-D-25-33411R1

Dear Dr. Qin,

We’re pleased to inform you that your manuscript has been judged scientifically suitable for publication and will be formally accepted for publication once it meets all outstanding technical requirements.

Kind regards,

Afzal Basha Shaik, Ph.D

Academic Editor

PLOS ONE
---

## [Editor Report · Acceptance letter]

PONE-D-25-33411R1

PLOS ONE

Dear Dr. Qin,

I'm pleased to inform you that your manuscript has been deemed suitable for publication in PLOS ONE. Congratulations! Your manuscript is now being handed over to our production team.

Kind regards,

on behalf of

Dr. Afzal Basha Shaik

Academic Editor

PLOS ONE